# Evaluating quality in adolescent mental health services: a systematic review

Meaghen Quinlan-Davidson  ,[1] Kathryn J Roberts,[1] Delan Devakumar,[1] Susan M. Sawyer,[2] Rafael Cortez,[3] Ligia Kiss[1]

► Prepublication history and additional online supplemental material for this paper are available online. To view these files, please visit the journal online (http://dx.doi.org/10.1136/bmjopen-2020-044929).

¹Institute for Global Health, University College London, London, UK
²Centre for Adolescent Health, Royal Children's Hospital; Murdoch Children's Research Institute; and Department of Paediatrics, The University of Melbourne, Parkville, Victoria, Australia
³Health, Nutrition and Population, The World Bank, Washington, District of Columbia, USA

**Correspondence to**
Meaghen Quinlan-Davidson; meaghen.quinlan-davidson.17@ucl.ac.uk

## ABSTRACT

**Objectives** To evaluate the quality of adolescent mental health service provision globally, according to the WHO Global Standards of adolescent mental health literacy, appropriate package of services and provider competencies.

**Design and data sources** Systematic review of 5 databases, and screening of eligible articles, from 1 January 2008 to 31 December 2020.

**Study eligibility criteria** We focused on quantitative and mixed-method studies that evaluated adolescent mental health literacy, appropriate package of services and provider competencies in mental health services, and that targeted depression, anxiety and post-traumatic stress disorder among adolescents (10–19 years). This included adolescents exposed to interventions or strategies within mental health services.

**Study appraisal and synthesis methods** Study quality was assessed using the National Institutes for Health Study Quality Assessment Tools. Data were extracted and grouped based on WHO quality Standards.

**Results** Of the 20 104 studies identified, 20 articles were included. The majority of studies came from high-income countries, with one from a low-income country. Most of the studies did not conceptualise quality. Results found that an online decision aid was evaluated to increase adolescent mental health literacy. Studies that targeted an appropriate package of services evaluated the quality of engagement between the therapist and adolescent, patient-centred communication, mental health service use, linkages to mental health services, health facility culture and intensive community treatment. Provider competencies focused on studies that evaluated confidence in managing and referring adolescents, collaboration between health facility levels, evidence-based practices and technology use.

**Conclusions and implications** There is limited evidence on quality measures in adolescent mental health services (as conforms to the WHO Global Standards), pointing to a global evidence gap for adolescent mental health services. There are several challenges to overcome, including a need to develop consensus on quality and methods to measure quality in mental health settings.

**PROSPERO registration number** CRD42020161318.

## INTRODUCTION

Globally, a significant burden of disease is due to mental health disorders.[1–3] Symptoms largely emerge during childhood, adolescence and young adulthood,[1–4] with

### Strengths and limitations of this study

► This is the first review to investigate quality measures for adolescent mental health services globally.
► This review highlights a critical gap in evidence on quality in adolescent mental health services.
► This review was limited to mental health services to adolescents in health facilities. It did not review the quality of mental health preventative or health promotion activities that are typically provided in other settings such as communities or schools.
► The review investigated adolescents with depression, anxiety and post-traumatic stress disorder and not those with other mental health conditions.

estimates suggesting that by 24 years of age, 75% of adult mental health disorders have appeared.[5 6] Current global estimates indicate that 14.8% of young people aged 10–24 years have a mental health disorder,[7] with the most common being unipolar depressive disorders, anxiety disorders and self-harm.[5] Depressive disorders rank as the fourth highest contributor to disability adjusted life years in 10–14 years old (4.3%) and the third highest among 15–19 years old (5.6%). Meanwhile, depressive and anxiety disorders rank among the top six contributors to years lived with disability for 10–14 years old (7.4% and 5.0%, respectively) and among the top five for 15–19 years old (11.2% and 5.8%, respectively).[3] In addition, a meta-analysis found that 16% of children and adolescents (2–18 years of age) develop post-traumatic stress disorder (PTSD) after exposure to a traumatic event.[8] Depression, anxiety and PTSD can be influenced by internal and external stressors.[9–11] These stressors can arise from living in challenging settings, experiencing traumatic events and life transitions,[10 12 13] and are exacerbated by the COVID-19 pandemic.[14 15] Despite their prevalence, the quality of mental health services, particularly within challenging environments, is limited.[9]

Mental health conditions have short-term and long-term health, education, social and

## Box 1 Health service barriers to quality adolescent mental health services

### Health service barriers
► Limited financial resources.
► Limited literacy on quality in mental healthcare, with limited or lack of quality criteria or measures.
► Lack of routine assessment of quality in services.
► Limited availability of specialised mental healthcare providers or those appropriately trained in mental health to engage and respond to adolescents' mental health needs.
► Healthcare provider stigma, as a behavioural barrier to quality mental healthcare.
► Lack of information, education and engagement from the health system to the community about adolescent mental health services and promotion of adolescent mental health literacy.
► Limited adolescent involvement in decisions about their treatment and care plans that are required for patient-centred care.
► Lack of youth-friendly, or patient-centred, services that ensure privacy, confidentiality (including confidentiality from parents and other service providers), respect, high-quality communication and services that are non-judgemental and free of embarrassment.
► Lack of equitable (geographically and socioeconomically) distribution of mental health services.
► Overburdened primary care systems which make it hard to integrate adolescent mental health services.
► Lack of mental health leadership from policy-makers and decision-makers at local and national government levels to champion quality in adolescent mental health services.

Sources: Knaak et al[96]; Saraceno et al[23]; McPherson[107]; Svirydzenka et al.[108]

economic impacts including early school leaving, social isolation, substance misuse, chronic unemployment and high economic costs for families, communities and health systems.[1–4] Yet, despite increased attention on adolescent mental health in recent years, adolescents with mental health conditions experience worse quality of care in comparison to other age groups.[16] Poor-quality healthcare is associated with worse health outcomes, comorbid conditions and a lack of trust in the health system. It also places the adolescent at an increased risk of death due to inadequate healthcare management, treatment and follow-up.[16]

Quality in mental healthcare is an important component of service delivery that contributes to adolescents seeking, receiving and continuing care. The Institute of Medicine and WHO define quality healthcare as 'the degree to which health services for individuals and populations increase the likelihood of desired health outcomes and are consistent with current professional knowledge'.[17] Adolescents benefit from mental health services that are adolescent-friendly, defined as accessible, acceptable, equitable, appropriate and effective.[18 19]

There are many multifaceted barriers that influence quality in mental healthcare services (box 1).

For adolescents and their families, barriers to quality mental healthcare start with access to these services. The stigma and discrimination associated with mental health

disorders, as well as how adolescents, families, communities and health services conceptualise emotional suffering and distress are significant barriers to access and use of quality mental health services.[20] Even in well-resourced contexts where mental health services are available, social and cultural conceptualisations of mental health influence the uptake and subsequent coverage of adolescent mental health services.[20] Adolescents are often less-experienced users of mental health services, with inadequate mental health literacy, including literacy about quality of care. They value privacy, confidentiality and patient-centred care, which includes respect, high-quality communication, and a therapist who is responsive to their needs. They seek to avoid judgement and embarrassment, and fear that their parents will be informed can limit their uptake of health services.[4]

There have been several global initiatives focusing on quality of healthcare in recent years (table 1). In 2015, the WHO developed Global standards for quality healthcare services for adolescents to support health service delivery and quality improvements for adolescents in primary and referral level facilities.[21] The aim of these standards was to increase adolescents' use of health services, improve health outcomes, ensure a minimal level of quality, and fulfil their rights to healthcare.[21] There are eight standards, all of which are important to ensuring quality in healthcare services. This includes adolescents' health literacy, community support, appropriate package of services, provider competencies, facility characteristics, equity and non-discrimination, data and quality improvement, and adolescent participation.[21] Notwithstanding the relevance of all standards to quality health services, three standards are particularly important for help-seeking behaviour among adolescents.[22–24]

1. *Adolescents' health literacy* whereby adolescents know about mental health and their own mental health (through the health facility), as well as knowing where health services are located and when to go.[21]
2. An *appropriate package of services* in which the health facility meets the needs of the adolescent by providing evidence-based information, counselling, correct diagnoses, treatment and care services.[21]
3. *Providers' competencies*, whereby healthcare providers are competent and provide effective care to adolescents (including respecting, protecting and fulfilling the rights of adolescents).[21]

These standards reflect the possibilities of interactions between adolescents and health services in terms of access, communication and competency of care.[16] Yet to date, there has been little research evaluating these standards with no systematic review of the evidence. Recent literature has argued that despite mental health conditions having their first onset during adolescence and young adulthood, these conditions often go undetected.[5 21 25 26] Adolescent mental health literacy empowers adolescents to recognise mental health symptoms and conditions, seek services, understand how they can improve their mental health, as well as combat stigma.[5 21 25] An appropriate

**Table 1** Definitions of quality by organisation and criteria

| Organisations | Definition | Criteria |
|---|---|---|
| Lancet Global Health Commission for High Quality Health Systems (2018)[16] | Quality health systems ensure that healthcare is optimised. This occurs by responding to population needs, providing care that maintains or improves health outcomes, and ensuring that all individuals feel valued | Quality framework components<br>► Quality impacts: better health, confidence in system and economic benefit<br>► Processes of care: competent care and systems, positive user experience<br>► Foundations: population, governance, platforms, workforce, tools |
| WHO, Organisation for Economic Co-operation and Development, World Bank (2018)[109] | How health services ensure desired health outcomes for the population, using evidence-based knowledge | Foundations critical to high quality health services:<br>► Healthcare workers: motivated and supported<br>► Healthcare facilities: accessible and well equipped<br>► Medicines/devices/technologies: safe in design and use<br>► Information systems: continual monitoring<br>► Financing mechanisms: enable and encourage quality |
| National Academies of Sciences (2018)[17] | | Six dimensions of quality<br>► Safety<br>► Effectiveness<br>► Person-centredness<br>► Accessibility, timeliness, Affordability<br>► Efficiency<br>► Equity |

package of services is key to overall quality of adolescent mental healthcare; it ensures that adolescents receive 'adolescent-friendly', comprehensive (promotion, prevention, diagnosis and treatment) mental healthcare. Prior evidence has found that health services for adolescents have focused on a limited range of services, such as sexual and reproductive health, with the service not equipped to deliver mental services to adolescents.[5 21] At the centre of providing quality adolescent mental healthcare is provider competencies, which includes providers' knowledge, attitudes and skills, as well as the provision of evidence-based care.[21 27 28] Prior evidence has found that healthcare providers often do not have the technical competence to promote, prevent and manage adolescent mental health cases.[4]

Despite a growing interest and investment in adolescent mental healthcare services, and the quality of such services, evidence on care quality and its effectiveness remain limited.[16] There is particularly poor information about how quality mental health services for adolescents should be developed and organised, clinicians trained, and health facility interventions implemented to improve adolescents' mental health outcomes and that meet their needs.[29–31] This gap is particularly relevant for adolescents in adverse family and social circumstances.[9 10] The objective of this systematic review is to evaluate the quality of adolescent mental health service provision globally, according to the WHO Global Standards of adolescent mental health literacy, provider competencies and an appropriate package of services.[21] This review focuses on quantitative and mixed-method evaluations of mental health services for 10-to-19 year olds adolescents with suspected or diagnosed cases of depression, anxiety and PTSD.

## METHODS

For this systematic review, we used the WHO Global Standards for quality healthcare services for adolescents, focusing on standards for mental health services.[21]

### Selection criteria

Inclusion criteria were quantitative and mixed-method studies that evaluated or assessed quality measures as defined by the WHO Global Standards (adolescent mental health literacy, provider competencies, and appropriate package of services) applied to existing mental health services targeted to adolescents (10–19 years) for depression, anxiety and PTSD. We used *the Lancet Global Health Commission for High Quality Health Systems*[16] conceptual framework to choose three of the WHO quality Standards. Although all dimensions of the conceptual framework are relevant to a high-quality health system, we were most interested in focusing on the population that the health service is serving, the delivery of competent mental healthcare and the systems and processes of providing care.[9] Depression and anxiety are the most common mental health outcomes in adolescents living in challenging environments.[32–37] PTSD is also associated with living in challenging environments. There is evidence that exposure to both interpersonal (eg, assault, war terrorism and injury due to violence) and non-interpersonal (eg, accidents, natural disasters, sudden death of a loved one, witnessing or hearing about death or death threats and life-threatening diseases) trauma, characteristics typical of challenging environments, is associated with the development of PTSD.[8 32–38] Furthermore, environments in which adolescents are more likely to experience adversities associated with these disorders are often in settings where quality mental healthcare is scarce.[9]

Articles that focused on adolescent mental health literacy, appropriate package of services, and provider competencies

were identified and classified according to the process and output criteria in the WHO report found in online supplemental material 2.[21] Articles were included if adolescents had used or were currently using mental health services, or were exposed to interventions or strategies within established mental health services. Mental health services were defined as health services delivered at the primary, secondary, or higher health facility level that offered prevention and treatment for anxiety, depression or PTSD, as well as any community-based initiatives originating from these health service levels. Articles about healthcare providers delivering mental health services to adolescents were also included. No exclusions were made by country. In cases where studies included individuals aged 18–24 years or 0–10 years, inclusion criteria were met as long as 10–19 years old were the primary population of the study, meaning>50% of the study population and the median being within the age range of interest.

While mental health services can be delivered through school-based health services, we restricted the search to more generic and specialised healthcare facilities. Transition services for adolescents to adult services were also excluded as we strictly focused on 10–19 years old. Clinical interventions that focused solely on improving health outcomes were also excluded if they did not detail how quality measures were used. Mental health conditions linked to a comorbid physical health condition or learning disability were also excluded, given the more complex ways in which depression and anxiety can present in these groups, and the extent to which PTSD can reflect comorbid physical conditions in particular. We did not include schizophrenia spectrum disorders, bipolar disorders or adolescent precursors of these two severe groups of mental health conditions, as the median age of onset is older than the 10–19 years old age range and these disorders are less common than anxiety, depression and PTSD. We did not include studies that focused on "at-risk mental states", as evidence indicates that current risk identification approaches are limited within mental health services.[39] Our primary interest was public health services, which led to us excluding private healthcare services. Family therapy was also excluded as we wanted to focus on services that more directly targeted adolescents.[4] Given the launch of the *Mental health Gap Action Programme* guidelines by the WHO in 2008[40] which highlights the large treatment gap globally, studies prior to 2008 were excluded.[41] No language restrictions were applied.

### Search strategy
The Preferred Reporting Items for Systematic Reviews and Meta-Analyses (PRISMA) methodology was used to select the articles (online supplemental material 3 contains the PRISMA checklist).[42] Peer-reviewed literature was searched through the following databases: Pubmed, PsycINFO, MEDLINE, EMBASE and LILACS from 1 January 2008 to 31 December 2020. Articles were also found by screening references in selected scientific articles that matched eligibility for inclusion. The search strategy is provided in table 2.

### Data collection and extraction
Titles and abstracts were exported to Endnote[43] and scanned for relevance by MQ-D. Articles were removed if they did not meet inclusion criteria or were duplicates. The full text of included articles was obtained and studies were classified in relation to: (1) mental health literacy; (2) appropriate package of services or (3) provider competencies. Dual screening was conducted by two authors (MQ-D and KJR) to ensure that they met the inclusion criteria for this review.

### Data synthesis and quality assessment
A narrative synthesis of the included studies was undertaken as the lack of homogeneity precluded a quantitative synthesis of findings. The methodological quality of the studies was assessed using the National Institutes for Health (NIH) Study Quality Assessment Tools.[44] Studies were assessed for sources of bias (eg, patient selection, performance, attrition and detection), confounding, study power and strength of causality in the association between interventions and outcomes.[44] Two reviewers (MQ-D and KJR) divided the studies in half (50/50) and rated each study independently. They then checked each other's coding for agreement. Disagreements were resolved through discussion with a third reviewer (LK). Based on the ratings of each component, each study received an overall rating of good, fair, poor. Extracted data were entered into a table of study characteristics, including the quality assessment ratings for each study (table 3 and additional information found in online supplemental material 4).

### Patient and public involvement
Patients and the public were not involved.

## RESULTS
### Study selection
Figure 1 shows the results of the search and selection strategy. Of 20 104 references identified, 456 full-text articles met inclusion criteria from which a total of 20 articles were included in the study.

| Table 2 | Search strategy of electronic databases | | |
|---|---|---|---|
| **Evaluation terms** | **Quality terms** | **Population** | **Setting** |
| Evaluation | Quality | Adolescent | Mental health service |
| Assessment | Health literacy | Youth | Primary mental health service |
| | Appropriate package of services | Teen | Mental health counselling |
| | Provider competencies | Young people | General practitioner service |

**Table 3** Characteristics of included studies

| Author and country | Type of mental health service | Study/evaluation design | Target population | Element(s) of quality addressed | Results | Quality assessment |
|---|---|---|---|---|---|---|
| Davidson, USA[45] | Secondary level services | Pilot randomised controlled trial (RCT) | n=18 providers n=32 children and adolescents (5–16 years; mean=11.5 years) | Appropriate package of services; Provider competency | Strong alliance between adolescents and providers, d=0.11; Assisted in skills-based learning, d=0.47; Adolescent satisfaction with treatment (Child/ Adolescent Satisfaction Questionnaire), d=0.53 | Fair |
| Jager, The Netherlands[60] | Secondary level services | Longitudinal prospective cohort | n=315 12–18 year olds (mean=15.2 years) | Appropriate package of services; Provider competencies | Smaller reduction of psychosocial problems when adolescents valued communication but didn't experience it (total difficulties score at T1 an average of 15.6 and at T2 13.9) | Fair |
| Ougrin, UK[56] | Tertiary level services | RCT | n=53 in intervention arm (0–17 years; mean=16 years); n=53 in control arm (0–17 years; mean=16 years) | Appropriate package of services | Lower occupied bed days between intervention and control arm (median 34 days, p=0.04); Intervention arm more effective and financially reasonable compared with control arm | Good |
| Dion, Canada[58] | Emergency department (ED) services | Cross-sectional surveyed | n=87 medical staff (nurses, residents, and physicians) | Provider competency | ED staff reported greater confidence in managing and referring patients based on years of employment (r=0.35, p<0.01); Physicians (83%) were more confident than nurses (8%) or residents (8%) (p=0.05); The majority of ED staff (67% of nurses, 64% of residents, and 83% of physicians) were satisfied with the programme | Fair |
| Spenser, Canada[59] | Tertiary level services | Cohort study | n=27 community paediatricians; n=16 outpatient mental health clinicians | Provider competency | Five mental health clinicians stated that having paediatricians on mental health team was positive; Over 20% of clinical activity for almost 12 paediatricians deals with mental health issues; seven mental health clinicians stated that the educational sessions led to increased knowledge about mental health of children and adolescents | Poor |
| Ayton, UK[57] | Secondary level services | Mixed methods including survey with consultants and review of case notes. | n=23 Child an Adolescent Mental Health Service (CAMHS) consultants surveyed; 33 case notes reviewed | Appropriate package of services; Provider competencies | Care Programme Approach (CPA) care plans audited assessed 96.4% of mental health and needs and 71.4% of family needs; other needs (housing: 33.9%, sexual health: 7.1%, and sign of relapse or crisis: 28.6%) minimally assessed in CPA plans | Fair |
| Aupont, USA[46] | Primary level services | Prospective cohort with adolescent service users | n=329 (m=12.3 years) | Appropriate package of services | The relative risks of staying in mental healthcare (instead of going back to paediatrics) was 7.5 for those with depression and 5.1 for those with anxiety disorders; The return rates to the referring paediatrician were 27.9% and 5.9% for adolescents with anxiety and major depressive disorders | Fair |

Continued

**Table 3** Continued

| Author and country | Type of mental health service | Study/evaluation design | Target population | Element(s) of quality addressed | Results | Quality assessment |
|---|---|---|---|---|---|---|
| Simmons, Australia[53] | Enhanced primary level services | Prospective cohort study | n=57, 12–25 years old used the decision aid and completed the postdecision assessment; n=48 completed the follow-up assessment (mean=18.5 years) | Mental health literacy; Provider competencies | 97% reported increased confidence in deciding about their own healthcare after using the decision aid (p=0.022); on shared decision making adolescents scored average of 37.4 (range 29–44); indicating high level of perceived involvement in the treatment process | Fair |
| Irvine, Ireland[62] | Community, primary, secondary and tertiary level services | Cross-sectional online survey | n=604, 11–21 years old | Mental health literacy, appropriate package of services, provider competencies | 73% of adolescents stated that they had been spoken to in a way that they could understand; 42% and 40% stated that they were given a choice in treatment/support and felt involved in their treatment decisions; general practitioner (GP) and ED services scored poorly across all quality indicators; community services scored the highest | Fair |
| Jager, the Netherlands[61] | Secondary level services | Longitudinal, prospective cohort study | n=211, 12–18 years old (mean=15.3 years) | Appropriate package of services; Provider competencies | Adolescents who valued patient centred communication but did not have their communications needs met were less likely to adhere to their treatment (OR: 2.8; 95% CI: 1.1 to 6.8) | Fair |
| Stevens, USA[47] | Tertiary level services | RCT | n=179, 11–20 years old (mean=17.2 years) | Appropriate package of services | No significant differences found between treatment and control arms | Good |
| Anderson, Australia[54] | Secondary level services | RCT | n=73, 12–18 year olds (mean=13.9 years) | Appropriate package of services | Adolescents in intervention (mean 5.77, SD 1.2) and control (mean 5.58, SD 1.34) reported strong working alliance; Adolescent working alliance was positively associated with compliance at 6 months follow-up (r=0.30, p<0.001) | Poor |
| Kapp, Switzerland[63] | Secondary level services | Cross-sectional study | n=663 patients 10+ years of age (mean=14 years) | Appropriate package of services | Patients who had time to formulate and ask questions had better alliance (p<0.001); Easy accessibility to CAMHS by phone had higher alliance scores (p=0.037) | Fair |
| Cairns, Australia[55] | Primary level services | Cohort | n=283 clinical charts of 12–25 years old (mean=18 years) | Appropriate package of services; Provider competencies | Emotional management and well-being goals were most frequently recorded; None of the analysed goals met criteria for being specific, measurable, and timed; 57% were specific while 14% were measurable; none had a timeframe; Goal quality was not associated with service retention | Fair |

**Table 3** Continued

| Author and country | Type of mental health service | Study/evaluation design | Target population | Element(s) of quality addressed | Results | Quality assessment |
|---|---|---|---|---|---|---|
| Ringle, USA[48] | Secondary level services | Medical record audit | n=727 medical records of 8–18 years old (mean=11.4 years) | Provider competencies | 46% of children and adolescents received care that was guideline concordant; Clients with worse functioning (OR=0.985, p<0.001), higher problem severity (OR=1.02, p=0.015), higher risk of harm to others (OR=1.61, p<0.001), more school problems (OR=1.48, p<0.001), and who had a diagnosis of depression (OR=1.37, p<0.05) or a conduct-related disorder (OR=1.37, p<0.05) at intake were more likely to receive less intensive services than those recommended by the guidelines. | Fair |
| Sattler, USA[49] | Mix of primary and secondary level | Medical record audit | n=694 medical records 7–17 year olds (mean=12 years) | Appropriate package of services; Provider competencies | Patients received 1.48 evaluations on average for psychiatric symptoms; 45.7% of all facilities used self-report measures and 5.2% used diagnostic interview; 23.2% of psychologists documented the use of diagnostic interviews, compared with 2% and 0% of psychiatrists and primary care physicians (p<0.01); 43% psychologists, 43.3% psychiatrists, and 39.3% of social workers more likely to document specific diagnoses compared with primary care (22.6%) (p<0.01) | Fair |
| Sattler, USA[50] | Primary and secondary level services | Medical record audit | n=801, 7–17 years old (mean=12.9 years) | Appropriate package of services; Provider competencies | 5.3% of anxiety disorder specialty clinics used structured diagnostic interviews; 21% of all health facilities used rating scales (28.9% of specialty clinics, 19.6% of general mental health clinics, and 15% primary care); Evaluations in specialised clinics resulted in specific diagnosis (p<0.001); rating scales were associated with specific diagnosis (p=0.04) | Fair |
| Higa-McMillan, USA[51] | Secondary level services | Medical record audit | n=2485 3–19 year olds (mean=13.2 years) | Appropriate package of services; Provider competencies | 55%–93% of cases use the following practices derived from the evidence base (PDEB) for adolescents: cognitive, psychoeducational, relaxation, modelling; 99.7% of youth had at least one PDEB over their treatment course | Fair |
| Rukundo, Uganda[64] | Primary, secondary and tertiary level | Clinical record review | n=50 providers | Appropriate package of services; Provider competencies | Since training: more children and adolescents had patient-centred assessments; decreased use of medication with more appropriate medication prescribed; increased use of psychological treatments; and non-CAMH professionals had greater diagnosis revisions and management of cases | Fair |

Continued

**Table 3** Continued

| Author and country | Type of mental health service | Study/evaluation design | Target population | Element(s) of quality addressed | Results | Quality assessment |
|---|---|---|---|---|---|---|
| Bardach, USA[52] | ED | Medical record review | n=22, 844 children and adolescents 6–17 years of age (majority 12–17 years) | Appropriate package of services; provider competencies | 62% and 82.3% of patients had follow-up within 7 days and 30 days, respectively; patients discharged from GPs and EDs were less likely to have follow-up compared with those discharged from psychiatric services; Follow-up within seven or 30 days of discharge was associated with an increased risk of a subsequent hospitalisation or ED visit for a mental health illness | Fair |

CAMHS, Child and adolescent mental health services ; CPA, Care programme approach; ED, emergency department; GP, General Practitioner; PDEB, Practices derived from the evidence base; RCT, randomized controlled trial.

## Study characteristics and settings

The majority of studies were conducted in high-income countries, namely the USA (eight studies),[45–52] Australia (three studies),[53–55] UK (two studies),[56 57] Canada (two studies),[58 59] the Netherlands (two studies),[60 61] Ireland[62] and Switzerland (one study).[63] One study was conducted in Uganda.[64] Most of the studies focused on evaluating or assessing the provision of services that are appropriate for adolescents (15 studies) and enhancing provider competencies (15 studies), with a minority focused on increasing adolescents' mental health literacy (two studies).

The services ranged from emergency,[52 58] to primary level[46 49 50 52 53 55 62 64]; secondary level mental health services[45 48–52 54 57 60–64] and tertiary level services.[47 52 56 59 62 64]

## Quality assessment

The majority of the studies were assessed as 'fair' quality (n=16, 80%); two (10%) studies were assed as 'good' quality and two (10%) were assessed as 'poor quality'. Studies rated as 'poor' mainly had lack of clarity about the methods and outcomes analysed, confounding and higher sources of bias. Ougrin *et al*[56] and Stevens *et al*[47] implemented studies that were rated as good quality.[47 56] Both conducted randomised controlled trials in tertiary level facilities. They also experienced low drop-out rates, high adherence to the interventions, and consistently used valid and reliable measures.[47 56]

## Conceptualising quality

The majority of studies did not conceptualise quality.[46 47 53 54 56–61 64] Where it was conceptualised, it was in reference to high quality care, defined as healthcare provider fidelity to evidence-based treatment models and adolescents' engagement in the treatment process (satisfaction and quality of engagement with therapists and adolescents)[45]; as quality indicators in terms of information and access, facilities and services and quality of care[62]; as quality indicators in child and adolescent mental health services, specifically around patient satisfaction and quality engagement between the therapist and adolescent[63]; or follow-up after hospitalisation for a mental illness.[52] Common themes within these studies were a focus on the processes of care and quality impacts (improved mental health and greater confidence of the health service and system).[16]

Other studies mentioned quality in relation to patient-centred communication or how providers adapt their communication style to meet the needs and preferences of their patients.[61] Quality was also considered in relation to goal setting between the therapist and adolescent patient and whether these goals were specific, measurable, achievable, realistic/relevant and timely.[55] The use of evidence-based assessments, practices and policies was another aspect of quality mentioned in some studies.[48–50 64] Other aspects of quality were linked to communication. These studies included: coordination of care between an in-patient mental health unit and a community service[57]; mental healthcare delivered through the emergency

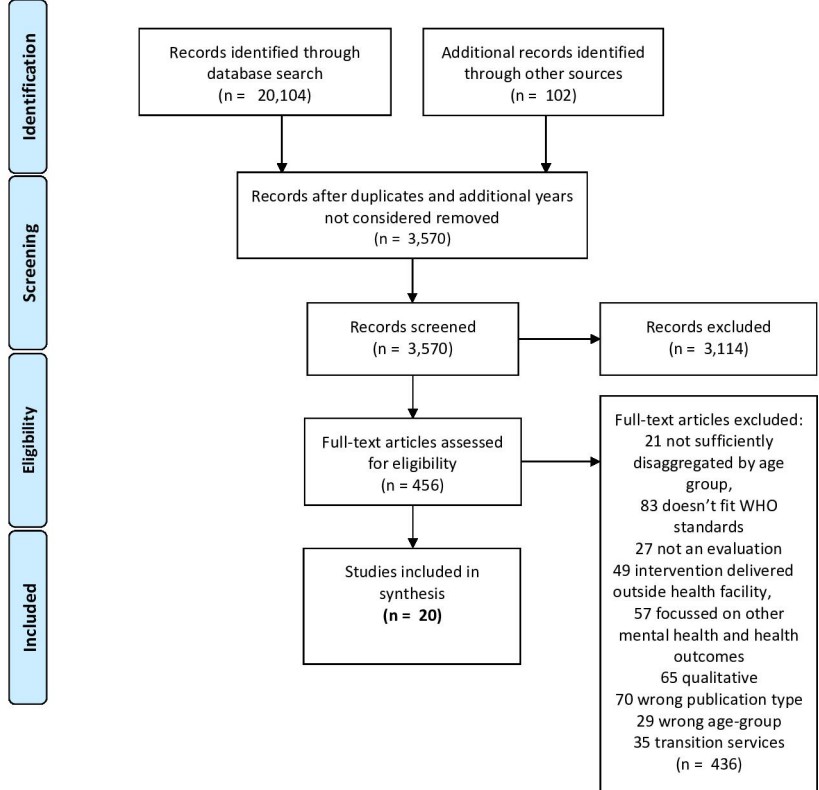

**Figure 1** Flow diagram of studies.

department (ED)[58] and collaboration between family physicians and psychiatrists.[59] Jager *et al*[60] measured affective quality as a key component of patient-centred communication, however, this was not defined.[60] The common theme that emerged from these studies is that they focused on foundations of quality care.[16] Findings from the review are presented below by WHO quality standard.[21]

## Quality Standards
### Adolescent Mental Health Literacy
Two studies reported outcomes relevant to adolescent mental health literacy. One focused on an online decision aid for mental health services.[53] Ninety-seven per cent of adolescents reported increased confidence and awareness in deciding about their own healthcare and involvement in the treatment process when exposed to an online decision aid (p=0.022).[53] Another evaluated adolescents' experience with mental health services; specifically, whether they were given useful information to understand their mental health needs and if they had a choice in their treatment and/or support.[62] EDs and general practitioner (GPs) scored poorly on these measures (<50% of adolescents experiencing this) while more than 50% of adolescents reported that they had experienced this in community Child an Adolescent Mental Health Service (CAMHS) and inpatient care.[62] The quality of evidence from these studies was fair.

### Appropriate Package of Services
Fifteen of the 20 studies evaluated services that met the WHO standard for appropriate packages of services.[21] Interventions that involved aspects of an appropriate package of mental health services were diverse and targeted the quality of engagement between the therapist and adolescent,[45 54 55 62 63] patient-centred communication,[60 61] mental health service use,[46 47 64] linkages to mental health services,[46] health facility culture and patient safety,[57] clinician's assessment of diagnostic and treatment services[49 50] and intensive community treatment.[56] The quality of the evidence for the 15 studies was poor to good.

Five of the 15 studies[45 54 55 62 63] reported improvements in the quality of engagement between therapist and adolescent patient, including the interaction, collaboration and bond.[54] These included the use of a tablet-based application on trauma-focused cognitive behavioural therapy (TF-CBT) in the USA,[45] an online CBT intervention in Australia,[54] an outpatient child and adolescent mental health clinic in Switzerland,[63] and the use of goal setting in Australia.[55] Davidson *et al*[45] evaluated a tablet-based application on TF-CBT, finding a small to medium effect size on developing therapeutic tasks (d=0.47) and a small effect size on therapeutic bond (d=0.11), with most adolescents satisfied with the intervention (d=0.53).[45] Anderson *et al*[54] evaluated an online CBT intervention with minimal therapist contact, finding that greater therapeutic alliance led to greater adolescent adherence

with treatment at 6-month follow-up (r=0.30, p<0.001). Results also found that adolescents in both the intervention and control arm reported strong therapeutic alliance.[54] Irvine[62] evaluated adolescents' experience at different types of health facilities; specifically, whether they felt involved in the decisions that were made about their treatment plan and if they found the support to be helpful[62]; 16%, 30%, 36% and 42% of adolescents stated that they felt involved in decisions about their care from in-patient facilities, EDs, community CAMHS and GPs, respectively, and 34%, 39%, 44% and 45% felt the support they received was helpful from EDs, in-patient care, GPs and community CAMHS, respectively.[62]

Two of the 15 studies[60 61] focused on patient-centred communication in the Netherlands, finding that adolescents who did not experience patient-centred communication were less likely to adhere (OR: 2.8, 95% CI 1.1 to 6.8) and have confidence (OR 4.5, 95% CI 1.8 to 11.6) in their course of treatment.[61] They also were less likely to experience a significant reduction in their mental health problems, compared with those who experienced patient-centred communication.[60]

One study evaluated mental health service use through a telephone support service (TSS) intervention in the USA.[47] The study found no difference in mental healthcare utilisation (p=0.65) between adolescents in the intervention and those in usual care.[47] Another study evaluated referrals between paediatric care to mental health services through the Targeted Child Psychiatric Service programme in the USA. The programme enables access to specialised mental health services for adolescents with mental health conditions from paediatric primary care, ensuring long-term management in the most appropriate healthcare setting. Results showed that adolescents with depression and anxiety required continued access to specialised mental healthcare.[46]

One study[57] evaluated health facility culture and patient safety with the Care Programme Approach (CPA), which ensures that children and adolescent patients are involved in all aspects of their mental healthcare in the UK. Patient safety was found to be an issue. In fact, unplanned discharge (ie, self-discharge, tribunal discharge or commissioning pressure to discharge) was the most common problem due to siloed rather than collaborative, decision making. Other challenges were limited collaboration between early intervention, education, and CAMHS teams; and a lack of joint protocols on the CPA and discharge between organisations.[57]

Two studies in the USA focused on the use of structured interviews, symptom rating scales, and Diagnostic and Statistical Manual of Mental Disorders diagnostic criteria in primary, secondary, and tertiary level services.[49 50] Results showed that structured diagnostic interviews were more likely to be used by psychiatrists and psychologists compared with general physicians (p<0.01).[49]

One study examined whether a supported discharge service (SDS), or intensive community treatment, would be more beneficial and cost-effective than usual care

among adolescents.[56] A significant difference in the overall number of bed-days from the SDS arm was found at 6 months (median 34 days, p=0.04) compared with usual care. The SDS was found to have at least 50% probability of being cost-effective in comparison to usual care and willingness to pay for outcome improvements (the incremental cost effectiveness ratio was −£991), or the cost per life year gained.[56]

### Providers' competencies

Fifteen studies included outcomes relevant to healthcare provider competency. They focused on confidence in managing and referring adolescents with mental health issues,[58] greater collaboration between paediatricians and mental health clinicians,[59] use of evidence-based practices,[48–52] use of a tablet to facilitate adolescent patient engagement in therapy,[45] provision of information[53 60–62] and implementation of care models and plans.[46 57 64] The quality of evidence for the fifteen included studies ranged from poor to fair.

In evaluating confidence of referring adolescents with mental health patients, Dion et al[58] found that training through a Crisis Intervention Programme increased ED staff confidence in managing and triaging patients (r=0.35, p<0.01) in Canada.[58] Another study evaluated collaboration between paediatricians and outpatient mental health clinicians in Canada. Results showed a positive effect on patient care with a paediatrician on the mental health team.[59]

Five studies evaluated the use of evidence-based practices and guidelines in adolescent mental health services.[48–51] Higa-McMillan et al[51] found that the most commonly used evidence-based practices for adolescents with anxiety disorders included cognitive, psychoeducational, relaxation and modelling.[51] In a USA study, the authors found that patients were more likely to receive less intensive services if they had poorer functioning, greater problem severity, greater risk of harm to others and greater school problems, with a diagnosis of depression or conduct disorder than guideline recommended.[48] As part of the Children's Core Set of quality measures, Bardach et al[52] evaluated the follow-up after hospitalisation for mental illness at 7 and 30 days for children and adolescents (aged 6–17 years old) in the USA. Results showed that 62% and 82.3% of patients were followed up within 7 and 30 days. Adolescent patients were more likely to be followed-up after discharge from psychiatric units and hospitals compared with those from general medical or surgical units.[52]

A 2-year child and adolescent mental health training programme for healthcare providers (psychiatric clinical officers, psychologists, psychiatric nurses, general nurses, occupational therapists, etc) was implemented in Uganda.[64] Medical records were reviewed annually over 6 years, finding that a greater number of children and adolescents were receiving thorough patient-centred assessments; a reduction in medication prescription; an increase in the use of psychological treatments and

greater management of cases by non-CAMHS professionals after the intervention.[64]

Two studies in the Netherlands found that adolescents who did not experience patient-centred communication were less likely to understand (OR: 3.7, 95% CI 1.5 to 9.0)[60 61] (OR: 3.1, 95% CI 1.1 to 8.5)[61] their course of treatment. Similarly, through the use of an online decision aid, 93% of participants were more likely to make a healthcare decision that was guideline concordant (p=0.004) and consistent with their preferences.[53]

## DISCUSSION

To our knowledge, this is the first review that has attempted to evaluate the quality of adolescent mental health services.[21] A total of 20 studies were identified, overwhelmingly from high-income countries. Fifteen studies focused on packages of services, 15 on healthcare provider competency and two on mental health literacy. There was limited evidence[53] of an intervention improving mental health, however, we cannot conclusively state this was effective.

Despite the large contribution of mental health conditions to the global burden of disease in adolescents and the need for quality mental healthcare services, we found that most studies lacked a formal conceptualisation of quality and did not have a clear framework or definition of quality. There were a variety of instruments used to measure quality and its indicators. Our understanding of quality in mental health services, as well as the generalisability of our findings, is therefore limited. Our findings also indicate a large service gap and suggests that there is a need to not only develop and standardise a definition of what constitutes quality adolescent mental healthcare, but also develop and standardise methods that measure quality in adolescent mental healthcare.

It should be noted that the WHO Global Standards were developed through a rigorous process.[41] This involved a needs assessment, the development of the Standards, consultations with experts, assessing the usability of the Standards through regional consultations and a country field test.[41] The WHO includes an implementation guide at the national, district and facility levels that identify actions needed to implement the Standards.[21] The majority of studies were conducted by universities in health facility settings. In reviewing the implementation guide, healthcare provider training and use of decision support tools at the health facility were the actions most relevant.[21] However, it was unclear the level of involvement of the health facility manager in the studies and whether there was an uptake of the intervention by the health facility after study completion. At the same time, there is a lack of peer-reviewed evidence on the implementation and evaluation of these standards, as well as a lack of specific and contextualised indicators to evaluate and monitor the Standards. This illustrates a gap between the literature and the Standards. The WHO (2015) does recognise that not all Standards will be implemented,

and that the standards were made to be evaluated and developed further once adapted and implemented at the national and regional levels.[41]

There are several challenges to providing quality care within adolescent mental health services, including stigmatising attitudes and behaviours about treatment seeking, service provision and utilisation,[65–68] the lack of professional expertise,[22–24] and the current disease-based model of medicine.[20] Also, quality in mental health services has received little attention in relationship to adolescents.[16] Indeed, stigma is a significant barrier to the availability and delivery of quality mental health services within communities.[13] It has been posited that stigma occurs at the structural (organisation, resources, quality standards), interpersonal (the quality of engagement between the healthcare provider and adolescent, patient safety) and intraindividual levels (healthcare providers unwilling to assess adolescent mental health conditions, adolescents unwilling to seek mental healthcare services).[69 70] This failure in quality prevents adolescents from seeking and continuing care for mental health conditions due to perceived stigma.[65–69 71] It points to a need for mental health literacy among healthcare providers. Quality mental health services cannot be achieved without healthcare providers having a reasonable understanding of adolescent mental health. This needs to start with healthcare provider training and preservice education.[71]

Apart from one study from Uganda,[47] all of the studies were from high-income countries, illustrating an important gap in the literature within low-income and-middle-income countries (LMICs) around quality of adolescent mental health services. This could reflect service and research gaps in all aspects of mental health in LMIC,[72–74] as well as different ways in which mental health is conceptualised in LMICs at the national and local levels.[20] It may also reflect the continued orientation in LMICs to more acute health conditions rather than in response to complex conditions that require long-term care, such as mental health conditions.[16 20] Patel and Saxena[20] argue that mental health conditions do not follow the typical disease-based model of medicine, and that a 'one size fits all' approach does not work.[20] This is particularly the case for subsyndromal or early onset mental health conditions which may not readily fit with diagnostically oriented services.[68]

There have been efforts to overcome these challenges, as identified in table 4. While the majority of these studies focused on generic healthcare for adolescents, their findings are equally relevant for adolescent mental healthcare.

Arguably, a good starting point for measuring quality in adolescent mental healthcare services would be a more scalable combination of the youth-friendly guideline driven care developed by Ambresin et al[28] and the quality standards developed by Sayal et al.[32] The same investments that promote quality in other age groups will be similarly valuable for adolescents, but greater specificity and focus is required around the health service aspects

**Table 4** Frameworks to address quality in adolescent health services

| Reference | Framework | Components of framework |
|---|---|---|
| Sawyer et al (2014)[110] | Conceptual framework for adolescent-friendly healthcare based on experience of care and evidence-informed care, including a set of 14 Indicators of quality healthcare for adolescents in hospitals | *Experience of care:*<br>Felt welcome in hospital<br>Age appropriate environment<br>Respected by clinicians<br>Trust in clinicians<br>Understanding of health information<br>Involvement in decisions about care or treatment<br>Comfort asking questions about health and well-being<br>*Evidence informed care:*<br>Psychosocial assessment<br>Confidentiality discussions<br>Time alone in consultations<br>Self-management<br>Transfer to adult services<br>Supported to continue education<br>Connection to external supports |
| Ambresin et al (2013)[28] | Domains for youth-friendly care to assess how well services are engaging young people | Accessibility of healthcare<br>Staff attitude<br>Communication<br>Medical competency<br>Guideline-driven care<br>Age-appropriate environment<br>Involvement in healthcare<br>Health outcomes |
| UK NHS (2007)[111] | 'You're Welcome' quality criteria to ensure that health services (primary, community, specialist and acute)are young people-friendly | Accessibility<br>Publicity<br>Confidentiality and consent<br>The environment<br>Staff training, skills, attitudes and values<br>Joined-up working<br>Monitoring and evaluation and involvement of young people<br>Health issues for adolescents<br>Sexual and reproductive health service<br>Child and adolescent mental health services |
| Sayal et al (2012)[31] | 10 quality standards for children and adolescents in primary mental healthcare | Confidentiality<br>Knowledge<br>Awareness<br>Communication<br>Continuity of care<br>Access and referral |

of engagement, communication and confidentiality. The studies identified in our review examined aspects of engagement and communication, but interestingly, did not explore confidentiality.

The findings from our review should be appreciated in light of the broader challenges to quality in adolescent mental health services as described above. Regarding adolescent mental health literacy, we found that a youth decision aid helped young people make evidence-informed decisions about their treatment, feel engaged in the process, and increased treatment adherence.[53] Previous literature has found similar results, with patients

reporting increased involvement in treatment decision making, increased knowledge about the treatment options and outcomes, and greater comfort making decisions.[75] As a foundation for quality, adolescents' knowledge shapes the way mental health services respond to them, and helps adolescents hold these services to account.[16] For packages of services, the quality of engagement between the adolescent and therapist was found to lead to positive outcomes.[76 77] The quality of the patient–therapist relationship has led to greater treatment efficacy,[78] increased autonomy, patient alliance and engagement, and greater favourable outcomes.[54 79] It ensures that the adolescent's perspective is included, that they consent and assent to their treatment plan,[80] and that they can address problems throughout the treatment process.[81] Evidence from the UK has shown that current services are not adequate for young people's mental health needs,[82–84] with youth reporting that they should be more engaged in the design of mental health services. From the studies on provider competency, we found that training general healthcare providers about adolescent mental health conditions helped build their confidence and knowledge when treating adolescents.[58 59 64] This aligns with previous literature as healthcare providers reported confidence, knowledge and a lack of specialised providers as barriers to care.[85 86] Provider competency is a foundation and process of care.[16] Adolescent mental healthcare providers require adequate clinical education and training on adolescent mental health. They also should provide evidence-based treatment, communicate clearly, ensure confidentiality and autonomy, promote timely and effective care and instil confidence in their adolescent patients that their conditions are being correctly detected and managed.[16]

The quality of evidence, assessed using the NIH Quality Assessment Tool, ranged from poor to good, and included various limitations in study design which could bias the results of the review. Furthermore, there was variation in the approach and tools used within evaluations, the content of the service, as well as the sample size. Certainly, within the three standards we systematically reviewed, it is not possible to identify the most effective standard of quality or service delivery method, as conforms to the WHO Global Standards.[21]

### Recommendations

We have several recommendations for further research on quality in adolescent mental health services to promote improved mental health outcomes. These recommendations are particularly important in light of the reported increase in adolescent mental health conditions associated with the COVID-19 pandemic.[87–91]

► First, to promote comparability, understanding and inform data collection, agreement around a developmentally-appropriate definition of quality would inform methods to measure quality in adolescent mental health services across different contexts.

► Second, collaborative research efforts, including the active participation of adolescents in this process,

are needed to strengthen the evidence on quality in adolescent mental health services, especially in LMICs. This includes research on adolescent mental health needs (particularly those exposed to daily adversity), research that articulates and tests the types of services that are best able to respond to their needs, knowledge of effective strategies to improve the quality of mental health services, including efforts to upskill the capabilities of all healthcare providers, not just mental health professionals, around adolescent mental health,[92] and evidence on the sustainability and effectiveness of what quality interventions to scale up through the health system, including psychotropic drugs and telehealth interventions. Furthermore, future studies should focus on psychotic disorders, which are also prevalent among adolescents and for which little evidence on quality of care is available.

► Third, health services need to proactively engage adolescents about their health needs, including mental health needs, and to ensure that they are informed about confidential services that are available to them, including vulnerable and at-risk adolescents.

► Fourth, to reduce stigma and close the well described treatment gap,[93–95] there needs to be greater investment in integrating adolescent mental health services into primary healthcare and training of non-specialised healthcare providers on adolescent mental health. This review shows that stigma negatively influences the quality of adolescent mental health services, which affirms the value of incorporating stigma reduction indicators into quality of care measures, as advocated by Knaak *et al*.[96] Cost-effectiveness analyses could help inform governments about the benefits to be gained when better mental health is reflected in higher school completion and regular employment.

### Study limitations

This review should be interpreted within the context of a number of limitations. We recognise that the WHO Global Standards quality framework is but one way of categorising quality. Leslie *et al*[97] warn that despite recent initiatives and greater focus on quality in healthcare, the various concepts and frameworks used to define and measure quality have led to 'inconsistent assessments and incomparable investments, leaving researchers and policymakers without direction'.[97] We recognise that beyond mental health services, the focus of this review, that there are various resources that can address adolescent mental health. This includes parenting interventions, which have been found to improve the mental health of adolescents[91] and school-based mental health interventions, which have also been found to contribute to improved health.[22] We also appreciate that an important aspect of quality care is continuity of care. For adolescents with persisting mental health issues, the transition from child or adolescent oriented services to adult oriented mental health services is a particularly important aspect of continuity of care. While this was beyond the scope of this review, it

is consistent with many adolescent services now using an extended definition of adolescence[98] up to 24 years.

None of the included studies focused on the quality of psychotropic drug prescriptions or the use of telehealth, as this was beyond the scope of the review and a limitation. Psychotropic drug prescriptions are one component of treatment for severe adolescent mental health conditions, with evidence from the USA in 2013 finding that 7% of adolescent participants were treated with psychotropic medication.[99] Despite this, high-quality evidence on the long-term effectiveness and safety of these medications for adolescents is limited, varying by condition and medication class.[99–101] The use of telehealth for mental health conditions has a long history, but has been growing in recent years as an accessible, efficient and cost-effective alternative to face-to-face consultations.[102] Evidence has found that telehealth is associated with patient satisfaction and is effective in evaluating and analysing mental health conditions.[102] In our review, we identified a number of studies that used different elements of telehealth. More recently, there has been appreciation of the particular benefits of telehealth within the context of the COVID-19 pandemic, including in low-resource settings. Evidence has found that it is associated with reduction in stigma and a higher participation rate among this age group, which may reflect adolescents ease with technology.[4 103–105] However, telehealth is also not without challenges, including around privacy, confidentiality, safety and equitable service use.[103 104] Limited access to the internet is a particular challenge to equitable telehealth. Telehealth sessions are typically easier to record which, in the context of informed consent, provides one mechanism to assess quality. Further research to identify which patients would most benefit from in-person visits or telehealth is indicated for common mental health conditions.[104 106]

## CONCLUSIONS

This review indicates the lack of consensus on quality in mental health services, with most of the identified studies failing to conceptualise quality at all. Many challenges remain around improving the quality of mental healthcare for adolescents.

**Correction notice** This article has been corrected since it was first published. The correct licence type is CC BY.

**Contributors** MQ-D led the study design, data searches, data extraction, quality appraisal, synthesis and drafted the manuscript. KJR contributed to the design, data extraction, quality appraisal and writing of the manuscript. LK and DD provided significant and critical contribution to the conceptualisation, design and interpretation of findings. SS critically reviewed the manuscript and offered key intellectual input to the synthesis and interpretation of findings. RC critically reviewed the manuscript and offered important intellectual input.

**Funding** This research received no specific grant from any funding agency in the public, commercial or not-for-profit sectors. KJR is supported by an Economic Social Research Council (ESRC) PhD studentship through the UBEL DTP (kr0001).

**Competing interests** None declared.

**Patient consent for publication** Not required.

**Provenance and peer review** Not commissioned; externally peer reviewed.

**Data availability statement** Data sharing not applicable as no datasets generated and/or analysed for this study. Not applicable.

**ORCID iD**
Meaghen Quinlan-Davidson http://orcid.org/0000-0002-7875-3753

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
