## [Reviewer comments · BMJ Open]

ARTICLE DETAILS

TITLE (PROVISIONAL)	Evaluating Quality in Adolescent Mental Health Services: A Systematic Review
AUTHORS	Quinlan-Davidson, Meaghen; Roberts, Kathryn; Devakumar, Delan; Sawyer, Susan; Cortez, Rafael; Kiss, Lúgia

VERSION 1 – REVIEW

REVIEWER	Giovanni de Girolamo St. John of God Clinical Rsearch Centre, Brescia (Italy)
REVIEW RETURNED	14-Dec-2020

GENERAL COMMENTS	This is an interesting paper discussing a very important topic (adolescent mental health). The paper is clearly written, and provides an interesting account of the scanty literature on this topic. There are some revisions to do, which may improve the readability and the usefulness of the paper. 1. On page 6 the authors state that they "used the Lancet Global Health..... conceptual framework" to choose three of the WHO quality standards. Actually, the WHO quality standards are eight: it is unclear WHY the authors have selected these three. They should provide a clear rationale.2. While it is true that two mental health conditions on which they have focused (e.g., diagnosed cases of depression and anxiety) are some of the most common mental health outcomes associated with living in challenging environments, this is less clear for PTSD. They should better justify the inclusion of PTSD among the conditions they selected.3. On the other hand, they have left out psychotic disorders. Although some comprehensive meta-analyses have shown that the mean age of onset for schizophrenia spectrum disorders and bipolar disorder is older than the age range they have selected (10-19 years), it is also true that in the last 20 years there has been a great deal of research on adolescent precursors of these two severe groups of mental disorders, and several landmark projects (e.g., Orygen in Australia, and others) have indeed targeted the all group of young people aged 15-25 years of age. Moreover, many studies have focused on at-risk mental states, involving many adolescents. Therefore, they should better justify the exclusion of this very important area of research and clinical practice.4. On page 30, line 21 please clarify the sentence " There was some evidence...".5. Table 3 "Characteristics of included studies" is incredibly long (12 pages!) and should be shortened; if necessary, the parts to be cut can be placed in a supplementary file.6. Box 2 Recommendations is too long: it should be shortened and changed in a list with bullet points.7. There are TWO main areas closely related to quality of mental
--

	health care, and they include the QUALITY OF PSYCHOTROPIC DRUG PRESCRIPTIONS (of utmost relevance in adolescence and young adulthood!!) and the USE OF TELEMENTAL HEALTH, universally recognized as an essential tool during the current pandemic. The authors should highlight the lack of any reference to these two areas of mental health care in the literature they have reviewed, and should make recommendations for the future.
--	--

REVIEWER	John Goodwin University College Cork National University of Ireland, School of Nursing and Midwifery
REVIEW RETURNED	08-Jan-2021

GENERAL COMMENTS	This is the foundation of an excellent systematic review, but there are some matters which need to be addressed. In particular, the search has not been updated in over a year. Introduction: Page 6, line 6: The following statement needs to be clarified. This includes adolescents' health literacy, community support, appropriate package of services, provider competencies, facility characteristics, equity and non-discrimination, data and quality improvement, and adolescent participation[14]. We used the Lancet Global Health Commission for High Quality Health Systems[9] conceptual framework to choose three of the WHO quality standards. We focussed on foundations of high-quality health systems, as adolescent knowledge helps define how health services respond to adolescents. We also focussed on competent care and systems within processes of care, or evidence-based, patient-centred care. How was the Lancet Global Health Commission for High Quality Health Systems' conceptual framework used to choose three of the quality standards? Were the other five standards not relevant? This needs to be much clearer. Methods Page 9, line 7: Private healthcare services and family therapy were also excluded. Why were these excluded? Page 9, line 17: Peer-reviewed literature was searched through the following databases: Pubmed, PsycINFO, MEDLINE, EMBASE, and LILACS from 1 January 2008 to 31 December 2019. Given that this is a full year ago, this search will need to be updated. Page 9, line 44: Titles and abstracts were exported to Endnote[27] and scanned for relevance. Was this conducted by two authors also?
---

	Page 10, line 8: The quality assessments were compared and disagreements resolved through discussion with a third reviewer. Was this always the same reviewer? Or did several reviewers resolve conflicts? Page 11: the layout of Table 3 needs to be adjusted (e.g. the “t” at the end of “quality assessment” should not appear on a line on its own Results Page 26, line 7: “Studies rated as ‘poor’ mainly had higher bias, lack of clarity on methods and outcomes analysed, and confounding”. Add “bias” after “confounding” Discussion Page 30, line 39. The following would be better placed under limitations We recognise that the WHO Global Standards quality framework is one way of categorising quality and we may have missed other frameworks. Leslie and colleagues (2018) warn that despite the recent initiatives and focus on quality in healthcare, the various concepts and frameworks used to define and measure quality leads to “inconsistent assessments and incomparable investments, leaving researchers and policymakers without direction”. Page 30, line 50: this paragraph is lacking references. It would also be better if this paragraph was combined with the following paragraph. Page 31, line 32: This is a “sentence fragment”: Particularly when some mental health conditions do not easily fit within diagnostic-led services and adolescents have subsyndromal conditions Page 32: Table 4: I recommend incorporating this information into the main text rather than having it side-lined from the discussion as a table. Page 35: It is unusual to see recommendations being addressed in a “box”, particularly a box this long. I suggest synthesizing this information and incorporating it into the main text.
--	--

VERSION 1 – AUTHOR RESPONSE

REVIEWER 1

1. This is an interesting paper discussing a very important topic (adolescent mental health). The paper is clearly written, and provides an interesting account of the scanty literature on this topic. There are some revisions to do, which may improve the readability and the usefulness of the paper.

Thank you.

2. On page 6 the authors state that they "used the Lancet Global Health..... conceptual framework" to choose three of the WHO quality standards. Actually, the WHO quality standards are eight: it is unclear WHY the authors have selected these three. They should provide a clear rationale.

Thank you. We recognise that all of the standards are important and relevant to ensuring quality health care services. We focussed on these three standards as prior literature has found that these specific standards facilitate help-seeking behaviour in adolescents and there's been little research to date on systematically evaluating these standards. In the revision, we have clarified this point on pages 6-7, providing greater justification of the context of this work.

Notwithstanding the relevance of all standards to quality health services, three standards are particularly important for help-seeking behaviour among adolescents.[22-24]

These standards reflect the possibilities of interactions between adolescents and health services in terms of access, communication and competency of care.[16] Yet to date, there has been little research evaluating these standards with no systematic review of the evidence. Recent literature has argued that despite mental health conditions having their first onset during adolescence and young adulthood, these conditions often go undetected.[5 21 25 26] Adolescent mental health literacy empowers adolescents to recognise mental health symptoms and conditions, seek services, understand how they can improve their mental health, as well as combat stigma.[5 21 25] An appropriate package of services is key to overall quality of adolescent mental health care; it ensures that adolescents receive "adolescent-friendly", comprehensive (promotion, prevention, diagnosis and treatment) mental health care. Prior evidence has found that health services for adolescents have focussed on a limited range of services, such as sexual and reproductive health, with the service not equipped to deliver mental services to adolescents.[5 21] At the centre of providing quality adolescent mental health care is provider competencies, which includes providers' knowledge, attitudes and skills, as well as the provision of evidence-based care.[21 27 28] Prior evidence has found that health care providers often do not have the technical competence to promote, prevent and manage adolescent mental health cases.[4]

3. While it is true that two mental health conditions on which they have focused (e.g., diagnosed cases of depression and anxiety) are some of the most common mental health outcomes associated with living in challenging environments, this is less clear for PTSD. They should better justify the inclusion of PTSD among the conditions they selected.

We have now expanded upon this point on page 9, providing evidence on how living in a challenging environment can lead to the development of PTSD.

Depression and anxiety are the most common mental health outcomes in adolescents living in challenging environments.[32-37] PTSD is also associated with living in challenging environments. There is evidence that exposure to both interpersonal (e.g., assault, war terrorism and injury due to violence) and non-interpersonal (e.g., accidents, natural disasters, sudden death of a loved one, witnessing or hearing about death or death threats and life threatening diseases) trauma, characteristics typical of challenging environments, is associated with the development of PTSD.[8 32-38] Furthermore, environments in which adolescents are more likely to experience adversities associated with these disorders are often in settings where quality mental health care is scarce.[9]

4. On the other hand, they have left out psychotic disorders. Although some comprehensive meta-analyses have shown that the mean age of onset for schizophrenia spectrum disorders and bipolar disorder is older than the age range they have selected (10-19 years), it is also true that in the

last 20 years there has been a great deal of research on adolescent precursors of these two severe groups of mental disorders, and several landmark projects (e.g., Orygen in Australia, and others) have indeed targeted the all group of young people aged 15-25 years of age. Moreover, many studies have focused on at-risk mental states, involving many adolescents. Therefore, they should better justify the exclusion of this very important area of research and clinical practice.

Thank you for your comments and suggestions. We did not include these conditions for the reason you provided, in terms of median age of onset, but also because these are less common than anxiety, depression and PTSD. We have added this information to page 9 in the manuscript.

We did not include schizophrenia spectrum disorders, bipolar disorders or adolescent precursors of these two severe groups of mental health conditions, as the median age of onset is older than the 10-19 year old age range and these disorders are less common than anxiety, depression and PTSD.

Although this was not the aim of our paper, we believe that studies on these disorder are important we included a recommendation in this sense:

Furthermore, future studies should focus on psychotic disorders, which are also prevalent among adolescents and for which little evidence on quality of care is available.

5. On page 30, line 21 please clarify the sentence “ There was some evidence...”.

Thank you for your suggestion. We have clarified this point on page 24 by revising the statement and including the one study that illustrated a change in mental health status based on the intervention.

There was limited evidence[45] of an intervention improving mental health, however, we cannot conclusively state this was effective.

6. Table 3 “Characteristics of included studies” is incredibly long (12 pages!) and should be shortened; if necessary, the parts to be cut can be placed in a supplementary file.

Thank you for your suggestion. We have revised Table 3 and included supplemental material 2 with additional information. Please see pages 12-17.

7. Box 2 Recommendations is too long: it should be shortened and changed in a list with bullet points.

We have shortened the recommendations. We also felt that we should incorporate the recommendations into the main text, as recommended by Reviewer 2. Please see page 28.

8. There are TWO main areas closely related to quality of mental health care, and they include the QUALITY OF PSYCHOTROPIC DRUG PRESCRIPTIONS (of utmost relevance in adolescence and young adulthood!!) and the USE OF TELEMENTAL HEALTH, universally recognized as an essential tool during the current pandemic. The authors should highlight the lack of any reference to these two areas of mental health care in the literature they have reviewed, and should make recommendations for the future.

Thank you for your suggestion. We agree that these are two important areas and have included these topics within the recommendations and limitations. Please see pages 28-29.

Recommendations

We also need evidence on the sustainability and effectiveness of quality interventions to scale up through the health system, including psychotropic drugs and telehealth interventions.

Limitations

None of the included studies focussed on the quality of psychotropic drug prescriptions or the use of telehealth, as this was beyond the scope of the review and a limitation. Psychotropic drug prescriptions are one component of treatment for severe adolescent mental health conditions, with evidence from the US in 2013 finding that 7% of adolescent participants were treated with psychotropic medication.[98] Despite this, high quality evidence on the long-term effectiveness and safety of these medications for adolescents is limited, varying by condition and medication class.[98-100] The use of telehealth for mental health conditions has a long history, but has been growing in recent years as an accessible, efficient, and cost-effective alternative to face-to-face consultations.[101] Evidence has found that telehealth is associated with patient satisfaction and is effective in evaluating and analysing mental health conditions. [101] . In our review, we identified a number of studies that used different elements of telehealth. More recently, there has been appreciation of the particular benefits of telehealth within the context of the COVID-19 pandemic, including in low-resource settings. Evidence has found that it is associated with reduction in stigma and a higher participation rate among this age group, which may reflect adolescents ease with technology.[4 102-104] However telehealth is also not without challenges, including around privacy, confidentiality, safety and equitable service use.[102 103] Limited access to the internet is a particular challenge to equitable telehealth. Telehealth sessions are typically easier to record which, in the context of informed consent, provides one mechanism to assess quality. Further research to identify which patients would most benefit from in-person visits or telehealth is indicated for common mental health conditions.[103 105]

REVIEWER 2

INTRODUCTION

1. This is the foundation of an excellent systematic review, but there are some matters which need to be addressed. In particular, the search has not been updated in over a year.

Thank you for your comments. We agree and have updated the review to include articles through 31 December, 2020. Please see pages 10 and 11.

The Preferred Reporting Items for Systematic Reviews and Meta-Analyses (PRISMA) methodology was used to select the articles.[42] Peer-reviewed literature was searched through the following databases: Pubmed, PsycINFO, MEDLINE, EMBASE, and LILACS from 1 January 2008 to 31 December 2020.

Figure 1 shows the results of the search and selection strategy. Of 20,104 references identified, 456 full-text articles met inclusion criteria from which a total of 20 articles were included in the study.

2. Page 6, line 6: The following statement needs to be clarified.

This includes adolescents' health literacy, community support, appropriate package of services, provider competencies, facility characteristics, equity and non-discrimination, data and quality

improvement, and adolescent participation[14]. We used the Lancet Global Health Commission for High Quality Health Systems[9] conceptual framework to choose three of the WHO quality standards. We focussed on foundations of high-quality health systems, as adolescent knowledge helps define how health services respond to adolescents. We also focussed on competent care and systems within processes of care, or evidence-based, patient-centred care.

How was the Lancet Global Health Commission for High Quality Health Systems' conceptual framework used to choose three of the quality standards? Were the other five standards not relevant? This needs to be much clearer.

Thank you, this is a good point. We agree and can see how this would be confusing to the reader. For the conceptual framework, we were most interested in investigating how health services are serving adolescents. As such, we focussed on foundations of a quality health care system. We also were interested in how evidence-based care is being implemented and focussed on processes of care. In terms of the WHO quality standards, we recognise that all of the standards are important and relevant to ensuring quality health care services. We focussed on these three standards as prior literature has found that these standards facilitate help-seeking behaviour in adolescents and there's been little research to date on evaluating these standards. We have clarified these points on pages 6-8.

Notwithstanding the relevance of all standards to quality health services, three standards are particularly important for help-seeking behaviour among adolescents.[22-24]

These standards reflect the possibilities of interactions between adolescents and health services in terms of access, communication and competency of care.[16] Yet to date, there has been little research evaluating these standards with no systematic review of the evidence. Recent literature has argued that despite mental health conditions having their first onset during adolescence and young adulthood, these conditions often go undetected.[5 21 25 26] Adolescent mental health literacy empowers adolescents to recognise mental health symptoms and conditions, seek services, understand how they can improve their mental health, as well as combat stigma.[5 21 25] An appropriate package of services is key to overall quality of adolescent mental health care; it ensures that adolescents receive "adolescent-friendly", comprehensive (promotion, prevention, diagnosis and treatment) mental health care. Prior evidence has found that health services for adolescents have focussed on a limited range of services, such as sexual and reproductive health, with the service not equipped to deliver mental services to adolescents.[5 21] At the centre of providing quality adolescent mental health care is provider competencies, which includes providers' knowledge, attitudes and skills, as well as the provision of evidence-based care.[21 27 28] Prior evidence has found that health care providers often do not have the technical competence to promote, prevent and manage adolescent mental health cases.[4]

We used the Lancet Global Health Commission for High Quality Health Systems[16] conceptual framework to choose three of the WHO quality Standards. Although all dimensions of the conceptual framework are relevant to a high-quality health system, we were most interested in focussing on the population that the health service is serving, the delivery of competent mental health care and systems and processes of providing care.[9]

METHODS

3. Page 9, line 7: Private healthcare services and family therapy were also excluded. Why were these excluded?

We have added to page 9, explaining why we excluded these private healthcare services. The scope of this review was only public services.

Our primary interest was public health services, which led to us excluding private healthcare services. Family therapy was also excluded as we wanted to focus on services that more directly targeted adolescents.[4]

4. Page 9, line 17: Peer-reviewed literature was searched through the following databases: Pubmed, PsycINFO, MEDLINE, EMBASE, and LILACS from 1 January 2008 to 31 December 2019. Given that this is a full year ago, this search will need to be updated.

Thank you for the suggestion. We have updated the search through 31 December 2020. Please see pages 9 and 11, with revisions made to the text on pages Please see pages 10-11 and table 3 on pages 12-17.

The Preferred Reporting Items for Systematic Reviews and Meta-Analyses (PRISMA) methodology was used to select the articles.[42] Peer-reviewed literature was searched through the following databases: Pubmed, PsycINFO, MEDLINE, EMBASE, and LILACS from 1 January 2008 to 31 December 2020.

Figure 1 shows the results of the search and selection strategy. Of 20,104 references identified, 456 full-text articles met inclusion criteria from which a total of 20 articles were included in the study.

5. Page 9, line 44: Titles and abstracts were exported to Endnote[27] and scanned for relevance. Was this conducted by two authors also?

The titles and abstracts were scanned by MQD; we have clarified this point on page 10.

6. Page 10, line 8: The quality assessments were compared and disagreements resolved through discussion with a third reviewer. Was this always the same reviewer? Or did several reviewers resolve conflicts?

Thank you for your question. This was always resolved with the same third reviewer (LK). We have clarified this point on page 10.

7. Page 11: the layout of Table 3 needs to be adjusted (e.g. the “t” at the end of “quality assessment” should not appear on a line on its own

We have revised the table, as recommended by Reviewer 1, and placed additional information in supplemental material 2. Please see pages 12-17.

RESULTS

8. Page 26, line 7: “Studies rated as ‘poor’ mainly had higher bias, lack of clarity on methods and outcomes analysed, and confounding”. Add “bias” after “confounding”

Thank you for your suggestion. We have revised this accordingly on page 19.

Studies rated as 'poor' mainly had lack of clarity on methods and outcomes analysed, confounding and higher sources of bias.

DISCUSSION

9. Page 30, line 39. The following would be better placed under limitations
We recognise that the WHO Global Standards quality framework is one way of categorising quality and we may have missed other frameworks. Leslie and colleagues (2018) warn that despite the recent initiatives and focus on quality in healthcare, the various concepts and frameworks used to define and measure quality leads to "inconsistent assessments and incomparable investments, leaving researchers and policymakers without direction".

Thank you for your suggestion. We have revised this accordingly and placed the text on pages 28-29 under "Study Limitations".

10. Page 30, line 50: this paragraph is lacking references. It would also be better if this paragraph was combined with the following paragraph.

Thank you for your comment. We have added references and combined the paragraph with the following. Please see page 24-25.

There are several challenges to providing quality care within adolescent mental health services, including stigmatising attitudes and behaviours about treatment seeking, service provision and utilisation, [65-68] the lack of professional expertise, [22-24] and the current disease-based model of medicine.[20] Also, quality in mental health services has received little attention in relationship to adolescents.[16] Indeed, stigma is a significant barrier to the availability and delivery of quality mental health services within communities.[13]

11. Page 31, line 32: This is a "sentence fragment": Particularly when some mental health conditions do not easily fit within diagnostic-led services and adolescents have subsyndromal conditions

Thank you for your comment. We agree and have adjusted this accordingly. Please see page 25.

This is particularly the case for subsyndromal or early onset mental health conditions which may not readily fit with diagnostically oriented services.[68]

12. Page 32: Table 4: I recommend incorporating this information into the main text rather than having it side-lined from the discussion as a table.

Thank you for your comment. We would like to keep the contents of Table 4 within table format due to the word count limitations. We hope you agree and understand our rationale for this.

13. Page 35: It is unusual to see recommendations being addressed in a "box", particularly a box this long. I suggest synthesizing this information and incorporating it into the main text.

Thank you for your suggestion. We agree and have incorporated it into the main text, synthesising it. Please see page 28.

VERSION 2 – REVIEW

REVIEWER	John Goodwin University College Cork National University of Ireland, School of Nursing and Midwifery
REVIEW RETURNED	24-Mar-2021
GENERAL COMMENTS	I am satisfied that my comments have been addressed - well done.